# Approximate Quantum State Preparation with Tree-Based Bayesian Optimization Surrogates

## Abstract

We study the problem of *approximate state preparation* on near-term quantum computers, where the goal is to construct a parameterized circuit that reproduces the output distribution of a target quantum state while minimizing resource overhead. This task is especially relevant for near-term algorithms where distributional matching suffices, but it is challenging due to stochastic outputs, limited circuit depth, and a high-dimensional, non-smooth parameter space. We propose CircuitTree, a surrogate-guided optimization framework based on Bayesian Optimization with tree-based models, which avoids the scalability and smoothness assumptions of Gaussian Process surrogates. Our framework introduces a structured layerwise decomposition strategy that partitions parameters into blocks aligned with variational circuit architecture, enabling distributed and sample-efficient optimization with theoretical convergence guarantees. Empirical evaluations on synthetic benchmarks and variational tasks validate our theoretical insights, showing that CircuitTree achieves low total variation distance and high fidelity while requiring significantly shallower circuits than existing approaches.

## 1 Introduction

Approximate state preparation is a core problem in quantum algorithm design, where the goal is to construct a low-depth, parameterized quantum circuit that reproduces the output distribution (measurement statistics) of a target quantum state while minimizing resource overhead (Amy et al., 2013a; Nam et al., 2018; Han et al., 2025). This task is especially critical in the context of near-term quantum hardware, which is constrained by short coherence times, limited gate fidelity, and strict circuit depth limits (Preskill, 2018; De Luca, 2022; Cowtan et al., 2020). Existing approaches often rely on domain-specific heuristics or gradient-based techniques (Smith et al., 2021; Younis et al., 2021; Khatri et al., 2019) that either do not scale to high-dimensional parameter spaces or assume access to analytic gradients, which may not exist for circuits evaluated through noisy quantum measurements (Murali et al., 2020). We therefore study approximate state preparation as a black-box optimization problem over a non-smooth, high-dimensional objective: the discrepancy between the output statistics of a parameterized circuit and those of a target state.

A natural approach is Bayesian Optimization (BO), which optimizes expensive black-box functions by constructing surrogate models (Brochu et al., 2010a; Snoek et al., 2012; Shahriari et al., 2015). However, standard BO methods typically employ Gaussian Process (GP) surrogates, which scale poorly and require smoothness assumptions that do not hold in quantum optimization problems (Wang et al., 2016). In addition, GPs do not naturally capture the bounded distributions arising from quantum measurements and often oversmooth the non-smooth loss landscape. To address this, we propose CircuitTree, a surrogate-guided approximate state preparation framework based on tree-based models following the spirit of (Han et al., 2021), specifically gradient-boosted regression trees (GBRTs), which are better suited for the high-dimensional and non-smooth optimization landscape induced by quantum circuit outputs (Head et al., 2021).

CircuitTree addresses approximate state preparation based on output statistics*, which is fundamental to tasks where measurement statistics, not full quantum amplitudes, determine success. Examples of use cases include variational cost estimation, generative quantum models, and amplitude encoding from classical datasets. These are precisely the near-term workloads limited by measurement access and noise; our goal is to target the novel area of distributional alignment for sampling- and

measurement-driven algorithms, rather than general-purpose, tomographic unitary, or state synthesis. We specifically show that these general-purpose methods are inefficient for our targeted tasks.

Beyond the surrogate choice, we introduce a structured decomposition of the parameter space that leverages the layered architecture of variational circuits (Holmes et al., 2022). This layerwise decomposition yields a principled form of block coordinate optimization: parameters within each layer are optimized in localized subspaces while synchronization across layers ensures global convergence. This structure enables distributed, sample-efficient optimization and improves stability relative to random partitioning. We formalize the surrogate-guided approximate state preparation problem and present theoretical guarantees under mild assumptions on noise stochasticity and model fidelity.

**Summary of our contributions:**

- We formulate approximate state preparation as a black-box optimization problem with structured parameter spaces and identify the challenges of standard BO in this setting.
- We propose a surrogate-guided framework, CircuitTree, using GBRT surrogates and introduce a scalable distributed subspace optimization strategy based on circuit structure.
- We provide convergence guarantees and analyze the impact of surrogate model fidelity and parameter decomposition on optimization performance for practical guidance.
- We empirically validate the framework on widely-used quantum benchmarks and variational tasks, showing that our method achieves low total variation distance and high fidelity with significantly shallower circuits than prior approaches.

## 2    PROBLEM SETUP

Let $\mathcal{U}$ denote the space of $n$-qubit unitary transformations parameterized by an angle vector $\boldsymbol{\theta} \in \Theta = [0, 2\pi)^d$. The goal of approximate state preparation is to find a parameterized quantum circuit $C(\boldsymbol{\theta})$ whose output distribution closely matches that of a target transformation $U^\star \in \mathcal{U}$ acting on a state $|\psi_0\rangle$ (Amy et al., 2013a; Nam et al., 2018; Han et al., 2025). $C(\boldsymbol{\theta})$ is constructed from a fixed ansatz $\mathcal{A}$ composed of $L$ layers of parameterized gates, such that $\boldsymbol{\theta}$ parameterizes the full circuit. Unlike full unitary synthesis, the target is not known analytically; it is only accessible through its action on a fixed input state $|\psi_0\rangle$ and the resulting measurement statistics (Murali et al., 2020; De Luca, 2022). This naturally formulates a black-box optimization problem (Luo et al., 2024b;a).

**Definition 2.1** (Approximate State Preparation Objective). *Given a target $U^\star$, input $|\psi_0\rangle$, and parametric circuit $C(\boldsymbol{\theta})$, the problem seeks*

$$\boldsymbol{\theta}^\star = \arg\min_{\boldsymbol{\theta} \in \Theta} \mathcal{L}(C(\boldsymbol{\theta}) |\psi_0\rangle, U^\star |\psi_0\rangle)$$

*where $\mathcal{L}$ measures the discrepancy between the output distributions induced by $C(\boldsymbol{\theta})$ and $U^\star$ on $|\psi_0\rangle$.*

The $U^\star$ in the above definition represents a conceptual reference transformation, not an executable circuit. Thus it does not need to be calculated direcly. The target distribution stemming from this $U^\star$, $p^\star$, can arise from classical computation (e.g., known analytic distribution for state preparation tasks in quantum machine learning) or prior measurement of a reference state, depending on the application.

We adopt the *total variation distance* (TVD) (Oh et al., 2024; Clark & Thapliyal, 2024; Patel & Tiwari, 2021) as the loss function $\mathcal{L}$, i.e., the $\ell_1$ distance between probability vectors. Let $p_{\boldsymbol{\theta}}$ and $p^\star$ denote the distributions obtained by measuring $C(\boldsymbol{\theta}) |\psi_0\rangle$ and $U^\star |\psi_0\rangle$ in the computational basis:

$$\mathcal{L}(C(\boldsymbol{\theta}) |\psi_0\rangle, U^\star |\psi_0\rangle) := \mathrm{TVD}(p_{\boldsymbol{\theta}}, p^\star) = \tfrac{1}{2} \sum_{x \in \{0,1\}^n} |p_{\boldsymbol{\theta}}(x) - p^\star(x)|.$$

Each query to $\mathcal{L}$ is stochastic, as it is estimated from a finite number of measurements (shots). Our implementation utilizes sample-based estimates of TVD from limited measurement shots, which ensure consistent convergence and avoid exponential costs, without would otherwise be incurred with full state tomography-based distances. Moreover, the objective is non-convex, non-differentiable, and highly non-smooth in general: small changes in $\boldsymbol{\theta}$ may propagate across layers and produce abrupt changes in output statistics (Preskill, 2018). This motivates surrogate models that can handle stochastic, discontinuous responses.

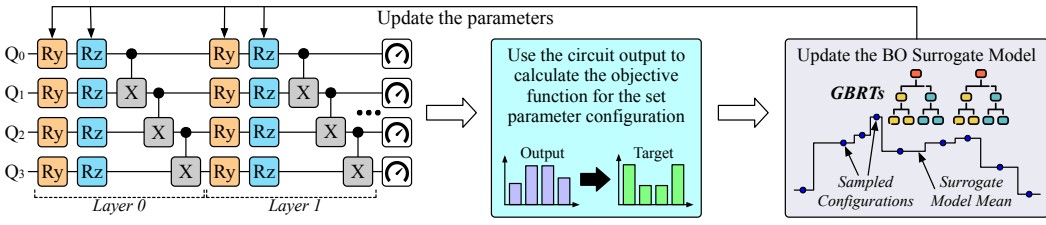

Figure 1: In this work, we use Bayesian Optimization (BO) with tree-based surrogates to update layered circuit parameters during approximate state preparation.

*Remark* 2.2. Unlike variational quantum algorithms, which optimize smooth cost functions derived from Hermitian observables, our objective arises directly from output distributions and is inherently non-smooth. This motivates the use of black-box optimization methods that do not rely on gradient information or smoothness (Shahriari et al., 2015; Luo et al., 2024b), and in particular surrogate models such as regression trees that naturally accommodate non-smoothness.

To optimize this objective, we employ surrogate modeling. Let $f(\boldsymbol{\theta}) := \mathcal{L}(C(\boldsymbol{\theta}), U^\star)$ denote the true cost. The surrogate $\hat{f}_t$ is a learned approximation trained on observed evaluations:

$$\mathcal{D}_t = \{(\boldsymbol{\theta}_i, y_i)\}_{i=1}^t, \quad y_i = f(\boldsymbol{\theta}_i) + \xi_i$$

where $\xi_i$ captures stochastic noise from measurement uncertainty or finite sampling. Evaluations on hardware may also include additional stochastic noise due to device imperfections such as thermal relaxation or depolarization (Patel et al., 2020b; Chakrabarti et al., 2019).

**Definition 2.3** (Surrogate-Guided Optimization). *At each round $t$, the optimizer fits $\hat{f}_t$ on $\mathcal{D}_t$ and selects the next query point via an acquisition function $\alpha_t : \Theta \to \mathbb{R}$:*

$$\boldsymbol{\theta}_{t+1} = \arg\max_{\boldsymbol{\theta} \in \Theta} \alpha_t(\boldsymbol{\theta}; \hat{f}_t).$$

The acquisition function balances exploration and exploitation; common examples include Expected Improvement (EI) and Upper Confidence Bound (UCB) (Brochu et al., 2010a; Shahriari et al., 2015). The goal is to minimize $f(\boldsymbol{\theta})$ with as few queries as possible, yielding a shallow circuit that approximates the target state's measurement statistics.

*Remark* 2.4. This problem departs from classical BO in key ways: (1) the function $f$ is distributional and highly non-smooth; (2) the parameter space $\Theta$ is structured by circuit layers and is only partially separable; and (3) the objective is defined relative to a fixed input state $|\psi_0\rangle$. These distinctions motivate both our surrogate choice (GBRT) and our structured layerwise optimization strategy.

## 3 SURROGATE MODELING AND OPTIMIZATION ALGORITHM

The core idea of our approach, named CircuitTree, is to learn a surrogate model that approximates the true state-preparation loss $f(\boldsymbol{\theta}) := \text{TVD}(p_{\boldsymbol{\theta}}, p^\star)$ and to use this surrogate to guide parameter updates (Fig. 1). Standard BO techniques often use Gaussian Process (GP) surrogates; however, GP-based models scale cubically with the number of observations, making them impractical in high-sample regimes (Nicoli et al., 2024; Benítez-Buenache & Portell-Montserrat, 2024). While GPs can be competitive for small datasets, they are also ill-suited for the highly non-smooth objectives that arise in approximate state preparation, such as minimizing TVD (Williams & Rasmussen, 2006).

Instead, we employ Gradient Boosted Regression Trees (GBRTs), an ensemble of tree-based learners that capture sharp discontinuities (Luo et al., 2024b; 2022) and perform well under limited data and at large scales (Nielsen & Chuang, 2010). Using an ensemble rather than a single tree also enables uncertainty quantification needed for acquisition functions. Compared to other tree-based methods, GBRTs also allow us to demonstrate theoretical convergence and empirical outperformance, as discussed further in this section and Sec. 6.

**Definition 3.1** (Surrogate Model). *A surrogate model $\hat{f}_t : \Theta \to \mathbb{R}$ is a regression function trained to approximate $f$ using dataset $\mathcal{D}_t$. In our framework, $\hat{f}_t$ is a GBRT model composed of $M$ decision trees, each trained sequentially on residuals of the previous stage.*

*At each boosting step, a regression tree $h_t(\boldsymbol{\theta})$ is fit to the negative gradient of the loss $\mathcal{L}$ evaluated at the current prediction $\hat{f}_{t-1}$:*

$$h_t = \arg\min_h \sum_{i=1}^{t-1} \left[ -\frac{\partial \mathcal{L}(y_i, \hat{f}_{t-1}(\boldsymbol{\theta}_i))}{\partial \hat{f}_{t-1}(\boldsymbol{\theta}_i)} \right] h(\boldsymbol{\theta}_i).$$

*The surrogate is updated by adding a scaled version of $h_t$:*

$$\hat{f}_t(\boldsymbol{\theta}) = \hat{f}_{t-1}(\boldsymbol{\theta}) + \nu \cdot h_t(\boldsymbol{\theta}),$$

*where $\nu$ is the learning rate controlling the contribution of each tree.*

### 3.1 ACQUISITION FUNCTION AND OPTIMIZATION STRATEGY

At each iteration $t$, the next query $\boldsymbol{\theta}_{t+1}$ is chosen by maximizing an acquisition function $\alpha_t : \Theta \to \mathbb{R}$ derived from the surrogate. We use the *expected improvement* (EI) criterion:

$$\alpha_t(\boldsymbol{\theta}) = \mathbb{E}\left[ \max(f_t^\star - \hat{f}_t(\boldsymbol{\theta}), 0) \right],$$

where $f_t^\star = \min_{i \leq t} y_i$ is the best observed value. In GBRTs, this expectation is approximated by quantile regression over ensemble predictions.

*Remark* 3.2. Unlike GPs, GBRTs do not natively provide posterior distributions. In CircuitTree, we estimate uncertainty by combining (i) quantile-based predictions and (ii) diversity across tree paths in the ensemble, following the approach of (Han et al., 2021; Meinshausen & Ridgeway, 2006). This empirical posterior enables our use of the EI or UCB-style acquisition functions.

### 3.2 LAYERWISE PARAMETER DECOMPOSITION

The parameter vector $\boldsymbol{\theta}$ is structured by circuit layers: each layer $\ell = 1, \ldots, L$ contains a subset $\boldsymbol{\theta}^{(\ell)}$. To exploit this structure, we introduce a distributed optimization strategy that partitions $\Theta$ into disjoint subspaces optimized independently, while others are fixed. See Appendix A for details.

**Definition 3.3** (Layerwise Decomposition)**.** *Let $\Theta = \Theta^{(1)} \times \Theta^{(2)} \times \cdots \times \Theta^{(L)}$. For each layer $\ell$, a local surrogate $\hat{f}_t^{(\ell)}$ is trained on $\mathcal{D}_t$ restricted to $\Theta^{(\ell)}$.*

This yields a principled block coordinate optimization: (1) each surrogate operates in reduced dimensionality, improving sample efficiency; (2) layers can be optimized in parallel; and (3) barren plateaus are mitigated by restricting updates to local subspaces (Holmes et al., 2022).

### 3.3 DISTRIBUTED SURROGATE-GUIDED OPTIMIZATION

Our full algorithm is presented in Algorithm 1. Each layer is optimized in parallel with periodic synchronization to ensure a globally consistent parameter set.

*Remark* 3.4. Layerwise decomposition with distributed surrogates improves sample efficiency, provides stability relative to random partitioning, and preserves convergence guarantees under mild assumptions. In addition, approximate state preparation only requires trusted reference statistics in some applications (e.g., VQE); in others, such as quantum signal processing with classical data, no reference is needed. When reference statistics are required, their cost can be amortized across repeated use of the prepared state for practical use.

---

**Algorithm 1** SURROGATEPREP($U^\star, |\psi_0\rangle, \mathcal{A}$)

1: Initialize $\boldsymbol{\theta}_0 \sim \text{Unif}(\Theta)$
2: Evaluate $y_0 = \text{TVD}(C(\boldsymbol{\theta}_0) |\psi_0\rangle, U^\star |\psi_0\rangle)$
3: Initialize dataset $\mathcal{D}_0 = \{(\boldsymbol{\theta}_0, y_0)\}$
4: **for** $t = 1$ to $T$ **do**
5:     Train GBRT surrogate $\hat{f}_t$ on $\mathcal{D}_{t-1}$
6:     **for** each layer $\ell = 1, \ldots, L$ **in parallel do**
7:         Fix all $\boldsymbol{\theta}^{(j)}$ for $j \neq \ell$
8:         Optimize $\alpha_t^{(\ell)}$ to get $\boldsymbol{\theta}_t^{(\ell)}$
9:         Evaluate $y_t^{(\ell)} = \text{TVD}(C(\boldsymbol{\theta}_t) |\psi_0\rangle, U^\star |\psi_0\rangle)$
10:       Update $\mathcal{D}_t \leftarrow \mathcal{D}_{t-1} \cup \{(\boldsymbol{\theta}_t, y_t^{(\ell)})\}$
11:     **end for**
12:     Synchronize $\boldsymbol{\theta}_t$ across layers
13: **end for**
14: **return** $\boldsymbol{\theta}_{\text{best}} = \arg\min_{(\boldsymbol{\theta}, y) \in \mathcal{D}_T} y$

---

# 4 THEORETICAL ANALYSIS

We now provide theoretical guarantees for the convergence of CircuitTree, our surrogate-guided approximate state preparation procedure. Our theoretical analysis builds upon standard proofs of Bayesian optimization convergence, with the novelty lying in adapting and proving these guarantees for non-Gaussian, tree-based surrogates under structured quantum parameter spaces, which, to our knowledge, is the first result of its kind. Below we present a condensed analysis; full details are given in Appendix B. We begin with assumptions on the cost function and surrogate model class.

**Assumption 4.1** (Lipschitz Continuity). *The true loss $f : \Theta \to \mathbb{R}$ is $L$-Lipschitz w.r.t. the $\ell_2$ norm:*

$$|f(\boldsymbol{\theta}) - f(\boldsymbol{\theta}')| \leq L\|\boldsymbol{\theta} - \boldsymbol{\theta}'\|_2 \quad \forall \boldsymbol{\theta}, \boldsymbol{\theta}' \in \Theta.$$

**Assumption 4.2** (Bounded, Centered Noise). *At step $t \geq 1$ the algorithm queries $\boldsymbol{\theta}_t$ and observes*

$$y_t = f(\boldsymbol{\theta}_t) + \xi_t, \qquad \mathbb{E}[\xi_t] = 0, \quad |\xi_t| \leq \sigma \text{ a.s.}$$

We note that $\mathbb{E}[\xi_t]$ here refers not to the expectation of the TVD absolute value but to the expectation of the stochastic estimator of $f(\theta)$ used in BO, ensuring bounded variance rather than unbiasedness. The unbiasedness is with respect to the stochastic noise term $\xi_t$.

**Assumption 4.3** (Variance Floor at Unexplored Points). *For any unobserved $\tilde{\boldsymbol{\theta}} \notin \{\boldsymbol{\theta}_i\}_{i=1}^t$, at least one tree assigns $\tilde{\boldsymbol{\theta}}$ to an empty leaf determined by covariate splits. Thus the ensemble predictive variance at $\tilde{\boldsymbol{\theta}}$ is bounded away from zero.*

*Remark* 4.4. Assumption 4.1 is standard in BO analyses (Shahriari et al., 2015); although $f$ is globally non-smooth, local Lipschitzness suffices for regret bounds. Assumption 4.2 is a simplification: while hardware noise is not strictly bounded, it is well-approximated by sub-Gaussian distributions with bounded variance due to error mitigation (Preskill, 2021; Patel et al., 2020b; Silver et al., 2023). Assumption 4.3 follows prior tree-based BO work (Luo et al., 2024b; Han et al., 2021), ensuring unexplored regions remain attractive under UCB/EI.

To establish convergence, we first need to guarantee that unexplored regions do not collapse to zero variance under the surrogate.

**Lemma 4.5** (Predictive Variance at Unexplored Points). *Suppose $\tilde{\boldsymbol{\theta}}$ has never been queried up to round $t$. Then the ensemble variance satisfies*

$$s_t^2(\tilde{\boldsymbol{\theta}}) \geq \eta,$$

*where $\eta > 0$ depends only on past evaluations and the shrinkage parameter $\nu$.*

Since unexplored regions remain attractive, the optimizer continues to spread queries throughout $\Theta$. We formalize this with the covering radius.

**Definition 4.6** (Covering Radius). *The covering radius after $t$ rounds is*

$$\rho_t := \sup_{\boldsymbol{\theta} \in \Theta} \min_{1 \leq i \leq t} \|\boldsymbol{\theta} - \boldsymbol{\theta}_i\|_2,$$

*the maximum distance from any $\boldsymbol{\theta} \in \Theta$ to its nearest sampled point.*

As the covering radius shrinks, every region of $\Theta$ is eventually explored. Combining this with Lipschitz continuity gives the main result.

**Theorem 4.7** (Convergence under Layered Distributed Optimization). *Under Assumptions 4.1–4.3, the sequence $\{\boldsymbol{\theta}_t\}_{t=1}^T$ produced by SURROGATEPREP (Algorithm 1) satisfies*

$$\limsup_{t \to \infty} \mathbb{E}[f(\boldsymbol{\theta}_t)] \leq f^\star + \sigma,$$

*where $f^\star = \inf_{\boldsymbol{\theta} \in \Theta} f(\boldsymbol{\theta})$. If $\sigma = 0$, then*

$$\lim_{t \to \infty} \mathbb{E}[f(\boldsymbol{\theta}_t)] = f^\star.$$

*The convergence rate is $\mathcal{O}(t^{-1/d})$, where $d$ is the parameter space dimension.*

Here, the expectation value is taken with respect to the surrogate's stochastic predictions and measurement noise.

## 5 DISCUSSION

**Surrogate Fidelity.** The accuracy of the surrogate model directly bounds the regret incurred at each iteration: lower surrogate error leads to tighter guarantees on expected improvement (Shahriari et al., 2015). Gaussian Processes assume smoothness and offer closed-form uncertainty estimates (Snoek et al., 2012; Williams & Rasmussen, 2006), which makes them effective in small-sample regimes but computationally prohibitive at scale due to cubic complexity in the number of evaluations (Nicoli et al., 2024; Benítez-Buenache & Portell-Montserrat, 2024). By contrast, tree-based surrogates such as GBRT (Head et al., 2021; Taieb et al., 2016) are agnostic to continuity and scale linearly with the number of samples, making them well suited for the non-smooth landscapes encountered in approximate state preparation. Our analysis highlights the importance of ensemble-based acquisition heuristics to compensate for the lack of analytic posteriors, as also studied in quantile-based surrogates (Meinshausen & Ridgeway, 2006).

**Structured Parameter Spaces.** Quantum circuits often follow layered, modular architectures (Nam et al., 2020; Smith et al., 2021), which induce a block structure in the parameter space. Our layerwise decomposition exploits this structure by reducing dimensionality at each step and enabling distributed surrogates, yielding a principled form of block coordinate optimization with convergence guarantees (Theorem 4.7). This aligns with prior results in distributed and multi-fidelity BO (Swersky et al., 2013; Kandasamy et al., 2015). Empirically, the structured updates also mitigate barren plateaus by focusing optimization on local subspaces (Holmes et al., 2022). The mitigation arises not from altering gradient magnitudes but from restricting the optimization subspace through layerwise decomposition, which reduces parameter coupling/correlation and stabilizes optimization (explored empirically in Sec. 6).

**Expressivity vs. Trainability.** Highly expressive ansätze may require large parameter sets to approximate a target distribution (Khatri et al., 2019; Holmes et al., 2022), but this increases dimensionality and degrades trainability. Layered decomposition offers a compromise: restricting updates to low-dimensional subspaces while preserving global convergence. This mirrors results in variational quantum learning (Cerezo et al., 2021), where expressivity often trades off against trainability due to barren plateaus. Our results suggest that architectural priors, parameter tying, and regularization can further improve trainability without sacrificing fidelity, consistent with recent advances in ML-inspired compilation (Silver et al., 2022; Wang et al., 2022).

**Application Domains.** Our surrogate-guided approach is motivated by several application domains where approximate, measurement-level preparation is not only sufficient but preferable. Many variational algorithms rely on low-depth proxy states to explore ansatz landscapes before committing to a deeper, hardware-aligned parameterization; in such settings, being able to cheaply approximate a target distribution provides a strong warm-start without incurring the overhead of exact synthesis Gilchrist et al. (2005); Han et al. (2025); DiBrita et al. (2025). Approximate preparation also directly supports classical-quantum data matching tasks, such as aligning classical datasets with quantum embeddings, shadow-matching objectives in QML pipelines, or reconstructing coarse target distributions for training hybrid models, where exact internal dynamics are irrelevant and only output-level fidelity matters Pires et al. (2024); Amin et al. (2022); Schatzki et al. (2021). These domains illustrate why approximate state preparation is an important problem.

**Trusted Reference Statistics.** Finally, we clarify that approximate state preparation does not universally require access to trusted reference statistics. In applications such as quantum signal processing or classical data embedding, the target distribution is classically known and incurs no additional cost. In tasks such as VQE, where reference statistics are required, they can be amortized across repeated use of the prepared state, making the approach practical for near-term workloads.

## 6 EXPERIMENTS

Our experimental methods are explained in detail in Appendix C. Below we briefly summarize the methodology for brevity. We aim to answer the following questions:

**Q1** How do different surrogate models (GP vs. GBRT vs. Quantile Regression Forests (QRF)) compare in convergence speed, fidelity of approximate state preparation, and robustness?

**Q2** What is the effect of layerwise distributed optimization on convergence time and stability?

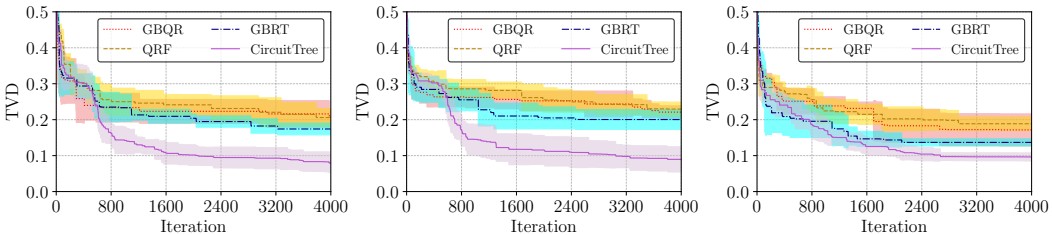

Figure 2: TVD during optimization of 3 different RQCs, using GBRT, QRF, and GBQR. GBRT significantly outperforms both QRF and GBQR in terms of TVD and runtime. CircuitTree's results with the final layered optimization design are shown for comparison.

**Q3** How query-efficient is CircuitTree in terms of quantum hardware measurements (shots), and how does it scale with ansatz depth and circuit width?

**Q4** Can CircuitTree reliably prepare application-relevant states, including those used in VQE and quantum linear algebra?

**Target Circuits.** We evaluate on three representative families of target states:

- **Random Quantum Circuits (RQC):** Circuits with randomly sampled gates (Boixo et al., 2018), used to test general-purpose approximate preparation.
- **Quantum State Preparation (QSP):** Amplitude-encoded states drawn from normalized Gaussian and uniform vectors used in ML applications (Schuld et al., 2019).
- **Variational Quantum Eigensolver (VQE):** Layered ansätze for estimating ground-state energies of Hamiltonians (Peruzzo et al., 2014).

**Ansatz.** We use a fixed layered ansatz consisting of parameterized $R_y$ and $R_z$ gates on each qubit for circuits of 4-8 qubits, followed by cascaded CX gates along a linear topology (e.g., 0–1, 1–2, 2–3). Each layer is repeated 3-4 times unless otherwise specified. The total number of parameters ranges from 24 to 32. We maintain this size not due to a technical limitation, as our technique can run circuits with tens of qubits. However, we found that beyond this size, the actual hardware results become essentially random due to the excessive noise inherent in current hardware. Thus, we choose these sizes to get meaningful output on today's hardware. The circuit gates are:

$$R_y(\theta) = \begin{pmatrix} \cos(\theta/2) & -\sin(\theta/2) \\ \sin(\theta/2) & \cos(\theta/2) \end{pmatrix}, \quad R_z(\lambda) = \begin{pmatrix} e^{-i\lambda/2} & 0 \\ 0 & e^{i\lambda/2} \end{pmatrix}, \quad CX = \begin{pmatrix} 1 & 0 & 0 & 0 \\ 0 & 1 & 0 & 0 \\ 0 & 0 & 0 & 1 \\ 0 & 0 & 1 & 0 \end{pmatrix}.$$

**Evaluation Metrics.** We report:

- **TVD:** Total Variation Distance between prepared and target output distributions.
- **Number of Shots:** Number of hardware measurements used during optimization.
- **Synthesis time:** Total classical runtime until convergence.
- **Circuit Depth and Gate Count:** Complexity of final ansatz instantiations.
- **Hardware Fidelity:** TVD between IBM hardware outputs and ideal simulation.

**Hardware and Runtime.** Experiments were run on AMD EPYC 7702P 64-core processors with x86_64 architecture and 2.0 GHz clock. Resource-bounded VMs of 8 cores, 32 GB memory, and 32 GB storage were used. Quantum evaluations were performed on IBM's `ibm_nazca`, a 127-qubit device (Eagle r3) (Castelvecchi, 2017) with median one-qubit gate error $3.34 \times 10^{-4}$, two-qubit error $1.15 \times 10^{-2}$, and measurement error $2.25 \times 10^{-2}$.

**Baselines.** We compare CircuitTree against BQSKit (Group, 2021), a leading synthesis toolkit. Unlike CircuitTree, which targets approximate state preparation under near-term constraints, BQSKit performs approximate general unitary synthesis using rule-based and numerical techniques. This baseline highlights the difference between generalized full circuit synthesis and the specialized approximate state synthesis problem addressed in this work.

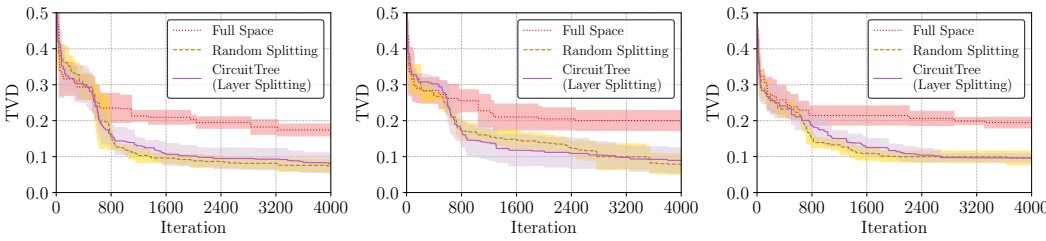

Figure 3: TVD during optimization of 3 different RQCs, using GBRT. Convergence is compared across full-space optimization, random subspace splitting, and layered splitting. CircuitTree adopts layered splitting with distributed surrogate optimization to maximize stability and fidelity.

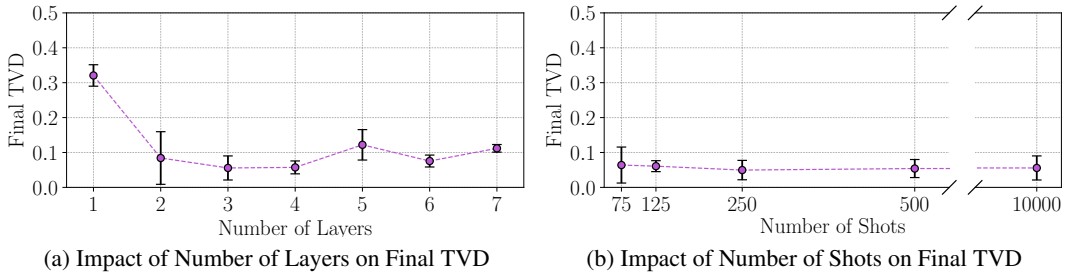

(a) Impact of Number of Layers on Final TVD      (b) Impact of Number of Shots on Final TVD

Figure 4: Analyzing the impact of (a) the number of ansätze layers and (b) the number of measurement shots on the performance of CircuitTree using VQE tasks.

## 6.1 SURROGATE COMPARISON (Q1)

We first compare surrogate models for guiding approximate state preparation. The candidates include: (1) Gaussian Processes (GP) (Duong et al., 2022; Benítez-Buenache & Portell-Montserrat, 2024; Nicoli et al., 2024), which provide probabilistic predictions and analytic uncertainty estimates but scale cubically in sample size and assume smoothness; (2) Gradient Boosted Quantile Regression (GBQR) (Taieb et al., 2016); and (3) Quantile Regression Forests (QRF) (Meinshausen & Ridgeway, 2006), both of which augment tree ensembles with explicit quantile modeling. All surrogates are embedded in the same Bayesian Optimization loop with Expected Improvement as the acquisition strategy. **Results.** Across three 3-layer RQCs, GBRT achieved the fastest convergence and lowest TVD (Fig. 2). **GP surrogates failed to finish within five days due to cubic scaling and the inability to capture sharp discontinuities.** QRF and GBQR offered quantile-based uncertainty but introduced runtime overhead without fidelity improvements. GBRT reached TVD $\leq 0.2$ with fewer evaluations and more than $2\times$ faster convergence, demonstrating robustness to non-smooth loss landscapes and practical suitability for near-term workloads.

## 6.2 LAYERWISE DISTRIBUTED OPTIMIZATION (Q2)

We next evaluate structured optimization strategies: (1) global surrogates trained over the full parameter space, (2) random subspace updates, and (3) our *layerwise distributed optimization*, which assigns each circuit layer an independent surrogate updated in parallel. **Results.** Fig. 3 shows that random subspaces improve over global surrogates but may introduce inconsistencies across layers, which may adversely affect convergence. Layerwise optimization achieved a $2.4\times$ reduction in convergence time and 50% lower final TVD. The advantage grows with deeper circuits, where synchronization overhead is outweighed by locality-aware updates. Independent per-layer surrogates allow meaningful improvements without incurring global coordination costs at every step. These findings empirically validate our theoretical results (Theorem 4.7) and highlight the importance of exploiting ansatz structure for efficient approximate state preparation.

## 6.3 HARDWARE-EFFICIENT SCALING (Q3)

We varied the number of ansatz layers (2–5) and the number of measurement shots (75 to 10,000) in the VQE preparation task to evaluate how these factors affect CircuitTree's fidelity. Each configuration

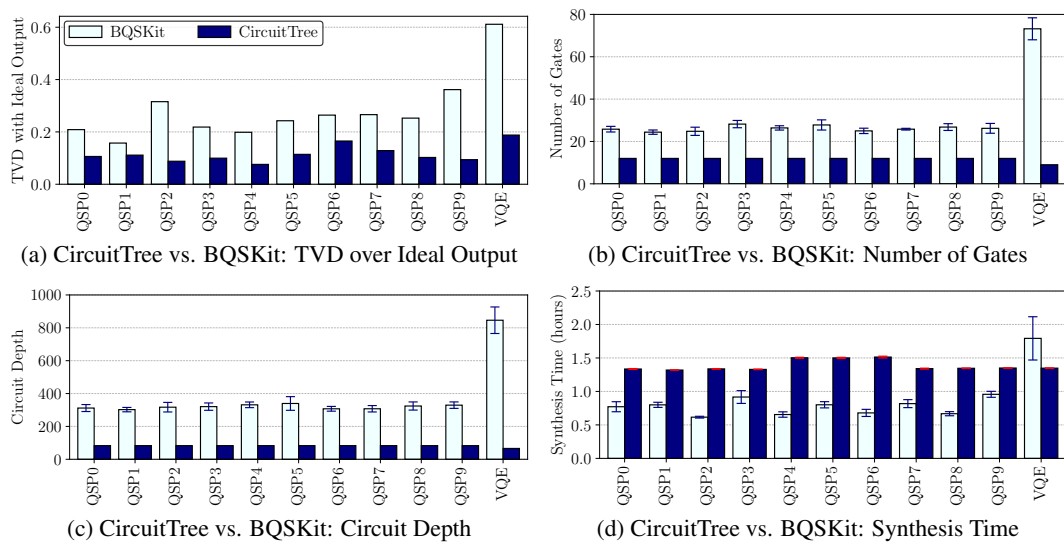

Figure 5: Comparison of CircuitTree and BQSKit on QSP and VQE workloads executed on IBM `ibm_nazca`. CircuitTree achieves higher fidelity with fewer gates and shallower depth, at the cost of increased but consistent classical runtime.

was repeated across multiple seeds to assess stability and convergence. **Results.** Fig. 4 shows that three to four layers yield the best balance between expressivity and trainability: CircuitTree consistently reached low TVD with minimal variance in this range. Two layers underfit the target distribution and exhibited unstable convergence, while five layers introduced over-parameterization that degraded performance. In terms of shot budget, 250 measurements were sufficient to achieve stable convergence. Using only 75 shots produced high variance and unreliable results, whereas increasing to 500 or even 10,000 shots offered no significant fidelity gains – a trend also observed for other circuits). These findings demonstrate that CircuitTree achieves hardware-efficient preparation with modest circuit depth and measurement overhead, making it well suited to near-term devices.

### 6.4 APPLICATION EVALUATION: QSP AND VQE (Q4)

We evaluate CircuitTree on application-relevant workloads executed on IBM's `ibm_nazca` quantum computer. We compare against BQSKit across three metrics: (1) fidelity measured as TVD between hardware and ideal simulation, (2) circuit complexity (depth and two-qubit gate count), and (3) synthesis runtime. **Results.** On real hardware, CircuitTree reduced output error by up to 59% compared to BQSKit (e.g., $0.12$ vs. $0.28$ for VQE), reduced two-qubit gate counts by 61%, and shortened circuit depth by 78%. These hardware-level improvements follow from surrogate-guided tuning of a fixed-depth ansatz, which ensures stable convergence and consistent circuit size. BQSKit, in contrast, produced variable-depth circuits with inconsistent fidelity. CircuitTree also exhibited lower variance across runs on QSP tasks ($0.02$ vs. $0.06$ standard deviation). While CircuitTree incurred approximately $1.5\times$ higher classical runtime, this overhead was predictable, purely classical, and offset by fidelity gains and noise robustness. Importantly, trusted reference statistics were only required for VQE tasks and could be amortized across repeated uses of the prepared state, making the approach practical for near-term workloads.

## 7 RELATED WORK

**Bayesian Optimization and Surrogate Modeling.** Bayesian Optimization (BO) is a standard framework for optimizing expensive black-box functions (Brochu et al., 2010b; Frazier, 2018). Classical BO typically employs Gaussian Process surrogates due to their closed-form posterior updates and uncertainty quantification (Snoek et al., 2012). However, GP-based methods scale poorly in high dimensions and rely on smoothness assumptions that break down in the discontinuous loss landscapes induced by quantum measurements (Wang et al., 2016). Scalable alternatives have been

proposed, including random forests (Hutter et al., 2011) and gradient-boosted trees (Head et al., 2021). Our contribution extends this line of work by analyzing tree-based surrogates in quantum state preparation and proving convergence guarantees under structured parameter spaces.

**Structured and Modular Optimization.** Decomposition strategies have been widely studied to improve sample efficiency in BO, including hierarchical models (Swersky et al., 2013), additive decompositions (Kandasamy et al., 2015; Patel et al., 2022), and factorized acquisition rules (Rolland et al., 2018). These often assume independence between subcomponents or rely on a known decomposition. In contrast, our approach exploits the explicit layered structure of quantum circuits to define distributed surrogate subproblems. This is related to block-coordinate descent and regional-division methods (Nesterov, 2012), but differs in that the global objective is never evaluated in full—only distributional statistics from layered surrogates guide optimization.

**Quantum Circuit Synthesis and State Preparation.** Traditional circuit synthesis relies on algebraic, rule-based, or template-matching approaches (Amy et al., 2013b; Nam et al., 2018; Smith et al., 2023; Paradis et al., 2024; Gidney et al., 2021; Kissinger et al., 2021; Yu et al., 2023; Miller et al., 2022; Nicoli et al., 2024; Tamiya & Yamasaki, 2022). More recent techniques include gradient-based variational optimization (Khatri et al., 2019) and probabilistic decomposition strategies (Group, 2021; Younis et al., 2021). These approaches typically assume access to gradients or explicit unitaries, both of which are impractical in near-term settings. Our work departs from this paradigm by framing approximate state preparation as a black-box optimization problem over distributional outputs, where gradients are unavailable and non-smoothness dominates.

Recent work has also explored complementary directions for circuit construction, particularly those targeting exact state reconstruction or tensor-network compilation. For example, Rudolph et al. employ tensor-network-based pretraining of parameterized circuits and also introduce a method for decomposing matrix product states into shallow quantum circuits (Rudolph et al., 2023b;a). Likewise, Shirakawa et al. propose an automated encoding strategy that produces a circuit capable of representing an arbitrary quantum state exactly (Shirakawa et al., 2024). These approaches aim for exact state construction or full tensor-network-based decomposition, both of which incur exponential scaling for general states. In contrast, our method focuses on approximate, measurement-driven state preparation and avoids full matrix or tensor-network decomposition entirely.

**Machine Learning for Quantum Compilation.** There is increasing interest in applying ML to quantum compilation, transpilation, and state preparation (Czarnik et al., 2021; Du et al., 2021; Zlokapa et al., 2023). Most existing methods are empirical and heuristic, offering limited theoretical foundations. By contrast, our contribution provides the first provable convergence guarantees for approximate state preparation using non-Gaussian surrogates, leveraging ensemble tree models and structured optimization to achieve both scalability and theoretical rigor.

## 8 CONCLUSION

We presented CircuitTree, a surrogate-guided framework for approximate quantum state preparation based on structured Bayesian Optimization. By combining tree-based surrogate models with a distributed, layerwise decomposition of the parameter space, our approach scales to high-dimensional, non-smooth objectives without relying on gradient information or full unitary access. We provided formal convergence guarantees under mild assumptions, and empirically validated the method's efficacy on both simulated and real hardware. Our results demonstrate that architectural structure in quantum circuits can be systematically exploited to improve surrogate-based optimization. More broadly, this work contributes to the growing intersection of structured black-box optimization and quantum algorithm design, showing that non-Gaussian surrogates with quantile-based uncertainty can deliver both scalability and provable convergence in near-term settings.

**Future Work.** Future directions include extending this framework to multi-input approximate preparation tasks and providing tighter analysis under stochastic and coherent noise models (non-zero $\sigma$). Beyond state preparation, we envision applying structured surrogate-guided optimization to broader classes of hybrid quantum–classical workloads where distributional fidelity, rather than full unitary synthesis, is the key metric of optimization. While our current focus is parameter optimization for fixed anszat, our method can also be combined with circuit-architecture search and exploration works to further enhance results by optimizing the ansatz sturcture.

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

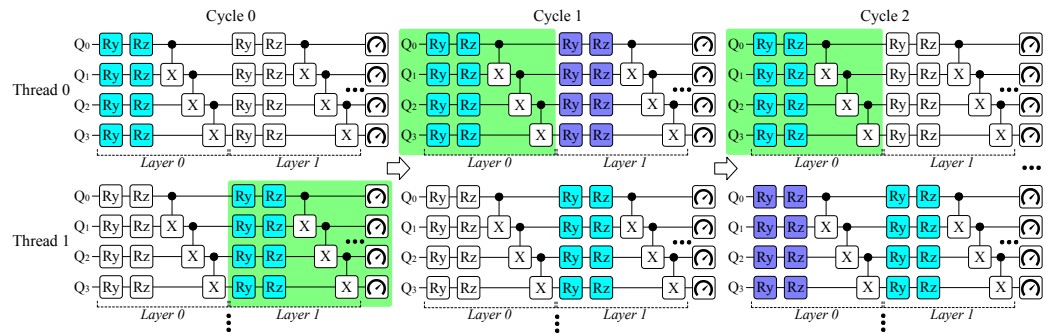

Figure 6: Distributed subspace splitting: each thread optimizes one layer (blue). Improvements over the prior cycle (green) are propagated to the global parameter vector (purple) used by all threads.

Mathias Weiden, Justin Kalloor, John Kubiatowicz, Ed Younis, and Costin Iancu. Wide quantum circuit optimization with topology aware synthesis. In *2022 IEEE/ACM Third International Workshop on Quantum Computing Software (QCS)*, pp. 1–11. IEEE, 2022.

Robert Wille, Lukas Burgholzer, and Alwin Zulehner. Mapping Quantum Circuits to IBM QX Architectures Using the Minimal Number of SWAP and H Operations. In *Proceedings of the 56th Annual Design Automation Conference 2019*, pp. 142. ACM, 2019.

Christopher KI Williams and Carl Edward Rasmussen. *Gaussian processes for machine learning*, volume 2. MIT press Cambridge, MA, 2006.

K Wright, KM Beck, S Debnath, JM Amini, Y Nam, N Grzesiak, J-S Chen, NC Pisenti, M Chmielewski, C Collins, et al. Benchmarking an 11-Qubit Quantum Computer. *Nature communications*, 10(1):1–6, 2019.

Ed Younis and Costin Iancu. Quantum circuit optimization and transpilation via parameterized circuit instantiation. In *2022 IEEE International Conference on Quantum Computing and Engineering (QCE)*, pp. 465–475. IEEE, 2022.

Ed Younis, Koushik Sen, Katherine Yelick, and Costin Iancu. Qfast: Conflating search and numerical optimization for scalable quantum circuit synthesis. In *2021 IEEE International Conference on Quantum Computing and Engineering (QCE)*, pp. 232–243. IEEE, 2021.

Wei Yu et al. Synthesis techniques for fault-tolerant quantum circuits. *Quantum*, 7:1–14, 2023.

Alex Zlokapa, Zoe Holmes, et al. Deep learning for quantum compilation. *Nature Machine Intelligence*, 5:449–456, 2023.

## A  DISTRIBUTED LAYERWISE OPTIMIZATION

### A.1  SUBSPACE SPLITTING

To reduce the dimensionality of the synthesis search space, we explored splitting the parameter vector into subspaces and alternating optimization across them. At each iteration, one subspace is optimized while the others are held fixed, ensuring all parameters are eventually tuned. We refer to this process as *subspace splitting*. A naive approach is to form subspaces by randomly grouping parameters. For the layered ansatz used in CircuitTree, however, splitting by layers is more natural: each subspace corresponds to the parameters of one circuit layer. This improves interpretability, since subspaces align directly with the gate execution order of the circuit.

## A.2 DISTRIBUTED SUBSPACE SPLITTING

While alternating subspaces improves trainability, we observed that the per-iteration progress within subspaces exceeded that of full-space optimization. To exploit this effect further, we developed a *distributed subspace* method: each subspace is assigned to a separate thread, which trains its own surrogate and optimizes concurrently. After an initial warm-up over the full parameter space, each thread runs for a fixed number of iterations in its assigned subspace. Whenever a thread achieves an improvement, it updates a shared global parameter vector that is then synchronized across all threads. This scheme is illustrated in Fig. 6. This distributed approach combines the benefits of subspace optimization with global consistency. As shown in Fig. 3, both random and layered splitting outperform full-space optimization, but layered splitting is preferred for CircuitTree due to its theoretical guarantees about convergence and alignment with circuit structure.

# B PROOF FOR THEOREM 4.7

Let

$$\Theta \subset \mathbb{R}^d, \qquad D = \operatorname{diam}(\Theta) < \infty, \qquad f : \Theta \to \mathbb{R}, \qquad f^\star = \inf_{\boldsymbol{\theta} \in \Theta} f(\boldsymbol{\theta}).$$

We restate the main assumptions for clarity.

**Assumption 4.1 (Lipschitz Continuity).** The loss function $f$ is $L$-Lipschitz with respect to the $\ell_2$ norm:

$$|f(\boldsymbol{\theta}) - f(\boldsymbol{\theta}')| \leq L\|\boldsymbol{\theta} - \boldsymbol{\theta}'\|_2 \quad \forall \boldsymbol{\theta}, \boldsymbol{\theta}' \in \Theta.$$

**Assumption 4.2 (Bounded, Centered Noise).** At step $t \geq 1$, the algorithm queries $\boldsymbol{\theta}_t$ and observes

$$y_t = f(\boldsymbol{\theta}_t) + \xi_t, \qquad \mathbb{E}[\xi_t] = 0, \quad |\xi_t| \leq \sigma \text{ almost surely.}$$

We note that $\mathbb{E}[\xi_t]$ here refers not to the expectation of the TVD absolute value but to the expectation of the stochastic estimator of $f(\theta)$ used in BO, ensuring bounded variance rather than unbiasedness. The unbiasedness is with respect to the stochastic noise term $\xi_t$.

A *GBRT* surrogate with $M_t$ trees is fitted at each round $t$. Let the $m^{\text{th}}$ tree be $h_m : \Theta \to \mathbb{R}$. With shrinkage parameter $0 < \nu \leq 1$,

$$\hat{f}_t(\boldsymbol{\theta}) = \sum_{m=1}^{M_t} \nu \, h_m(\boldsymbol{\theta}).$$

Define the empirical mean and variance across trees as

$$\mu_t(\boldsymbol{\theta}) = \frac{1}{M_t} \sum_{m=1}^{M_t} h_m(\boldsymbol{\theta}), \quad s_t^2(\boldsymbol{\theta}) = \frac{1}{M_t} \sum_{m=1}^{M_t} (h_m(\boldsymbol{\theta}) - \mu_t(\boldsymbol{\theta}))^2.$$

The algorithm selects query points via the UCB acquisition rule:

$$\boldsymbol{\theta}_{t+1} = \arg\min_{\boldsymbol{\theta} \in \Theta} \Big\{ \mu_t(\boldsymbol{\theta}) - \kappa_t s_t(\boldsymbol{\theta}) \Big\}, \tag{1}$$

where $\kappa_t > 0$ is an exploration multiplier. Empirically, we use expected improvement (EI), which is equivalent to Equation 1 for $\kappa_t = 1$. We analyze UCB for algebraic simplicity.

**Definition B.1** (Covering Radius). *The covering radius at round $t$ is*

$$\rho_t := \sup_{\boldsymbol{\theta} \in \Theta} \min_{1 \leq i \leq t} \|\boldsymbol{\theta} - \boldsymbol{\theta}_i\|_2.$$

## B.1 LOWER BOUND ON ENSEMBLE VARIANCE AT UNEXPLORED POINTS

**Lemma B.2.** *Fix $t \geq 1$. Suppose $\tilde{\boldsymbol{\theta}} \in \Theta$ has never been queried, i.e. $\tilde{\boldsymbol{\theta}} \neq \boldsymbol{\theta}_i$ for all $1 \leq i \leq t$. Assume that at least one tree assigns $\tilde{\boldsymbol{\theta}}$ to an empty leaf, i.e. a region of the partition containing no training points. Then the ensemble variance satisfies*

$$s_t^2(\tilde{\boldsymbol{\theta}}) \geq \eta,$$

*where*

$$\eta = \frac{\nu^2}{M_{\max}} \cdot \frac{1}{t} \sum_{i=1}^{t} (y_i - \bar{y}_t)^2 > 0, \qquad \bar{y}_t = \frac{1}{t} \sum_{i=1}^{t} y_i, \quad M_{\max} = \sup_{u \le t} M_u.$$

**Proof.**

1. For each tree $h_m$, let $\ell_m(\boldsymbol{\theta})$ denote the leaf containing $\boldsymbol{\theta}$. Define $A_m(\boldsymbol{\theta}) = \{i \le t : \boldsymbol{\theta}_i \in \ell_m(\boldsymbol{\theta})\}$, i.e. the indices of training points in the same leaf. The leaf prediction is

$$h_m(\boldsymbol{\theta}) = \frac{1}{|A_m(\boldsymbol{\theta})|} \sum_{i \in A_m(\boldsymbol{\theta})} r_{m,i},$$

   where $r_{m,i}$ is the residual for sample $i$ at tree $m$.

2. Square-loss boosting fits each tree $h_m$ to residuals $r_{m,i} = y_i - \hat{f}_{m-1}(\boldsymbol{\theta}_i)$. Least-squares fitting ensures

$$\frac{1}{t} \sum_{i=1}^{t} r_{m^\star,i}^2 = \min_c \frac{1}{t} \sum_{i=1}^{t} (r_{m^\star,i} - c)^2.$$

   Taking $c = \bar{r}_{m^\star} = \frac{1}{t} \sum_i r_{m^\star,i}$ yields

$$s_{\text{res}}^2 := \frac{1}{t} \sum_{i=1}^{t} (r_{m^\star,i} - \bar{r}_{m^\star})^2 > 0,$$

   since residuals cannot all be identical under bounded but varying noise.

3. For the tree $m^\star$ with an empty leaf at $\tilde{\boldsymbol{\theta}}$, we have $h_{m^\star}(\tilde{\boldsymbol{\theta}}) = 0$. Using variance decomposition across trees,

$$s_t^2(\tilde{\boldsymbol{\theta}}) \ge \frac{\nu^2}{M_t} s_{\text{res}}^2.$$

   Since $M_t \le M_{\max}$, we conclude

$$s_t^2(\tilde{\boldsymbol{\theta}}) \ge \frac{\nu^2}{M_{\max}} \cdot \frac{1}{t} \sum_{i=1}^{t} (y_i - \bar{y}_t)^2 = \eta > 0.$$

$\square$

### B.2 DENSITY OF QUERIES IN $\Theta$

**Lemma B.3.** *Fix $r > 0$ and $\tilde{\boldsymbol{\theta}} \in \Theta$. There exists a finite index $t_r(\tilde{\boldsymbol{\theta}})$ such that*

$$\|\boldsymbol{\theta}_{t_r(\tilde{\boldsymbol{\theta}})} - \tilde{\boldsymbol{\theta}}\|_2 \le r.$$

**Proof.** Let $B(\tilde{\boldsymbol{\theta}}, r)$ denote the open $r$-ball around $\tilde{\boldsymbol{\theta}}$. Suppose, for contradiction, that no $\boldsymbol{\theta}_i$ with $i \le t$ lies in $B(\tilde{\boldsymbol{\theta}}, r)$. Then Lemma 1 implies $s_{i-1}(\boldsymbol{\theta}) \ge \eta$ for all $\boldsymbol{\theta} \in B(\tilde{\boldsymbol{\theta}}, r)$.

Since $s_{i-1}(\boldsymbol{\theta}_{i-1}) \to 0$ as more points are sampled, choose $\kappa_{i-1}$ large enough that

$$\mu_{i-1}(\boldsymbol{\theta}_{i-1}) - \kappa_{i-1} s_{i-1}(\boldsymbol{\theta}_{i-1}) > \inf_{\boldsymbol{\theta} \in B(\tilde{\boldsymbol{\theta}}, r)} \left\{ \mu_{i-1}(\boldsymbol{\theta}) - \kappa_{i-1} s_{i-1}(\boldsymbol{\theta}) \right\}.$$

By the UCB rule in Equation 1, the next query point lies in $B(\tilde{\boldsymbol{\theta}}, r)$, contradicting the assumption. Hence such a $t_r(\tilde{\boldsymbol{\theta}})$ exists. Applying a Borel–Cantelli argument to a countable basis of rational balls implies $\lim_{t \to \infty} \rho_t = 0$, i.e. the query sequence is dense in $\Theta$. $\square$

### B.3 GEOMETRIC DECAY OF THE COVERING RADIUS

Let $C_d = \pi^{d/2}/\Gamma(1+d/2)$ be the volume of the unit ball in $\mathbb{R}^d$. A classical sphere-packing argument gives

$$\rho_t \leq \left(\frac{C_d D^d}{t}\right)^{1/d}, \qquad t \geq 1. \tag{2}$$

**Definition B.4** (Simple Regret). *The instantaneous simple regret is*
$$r_t = f(\boldsymbol{\theta}_t) - f^\star.$$

Since $\rho_{t-1}$ is the maximum distance to the nearest sampled point, there exists $i(t) \leq t - 1$ with
$$\|\boldsymbol{\theta}_t - \boldsymbol{\theta}_{i(t)}\|_2 \leq \rho_{t-1}.$$

Using Lipschitz continuity,
$$f(\boldsymbol{\theta}_t) \leq f(\boldsymbol{\theta}_{i(t)}) + L\rho_{t-1} \leq f^\star + L\rho_{t-1}. \tag{3}$$

Thus
$$\mathbb{E}[r_t] \leq L\rho_{t-1} + \sigma.$$

Substituting the geometric estimate Equation 2 into Equation 3 yields
$$\mathbb{E}[r_t] \leq L\left(C_d D^d\right)^{1/d} t^{-1/d} + \sigma, \qquad t \geq 1. \tag{4}$$

Summing from $t = 1$ to $T$ and comparing with $\int x^{-1/d} dx$, we obtain
$$\mathbb{E}\left[\sum_{t=1}^{T} r_t\right] = \begin{cases} \mathcal{O}(T^{1-1/d}) + \sigma T, & d > 1, \\ \mathcal{O}(\log T) + \sigma T, & d = 1. \end{cases}$$

Hence, in the noise-free case $\sigma = 0$, CircuitTree with UCB acquisition is a no-regret algorithm.

*Remark B.5.* For exploration, it suffices that $\kappa_t \to \infty$ while $\kappa_t s_t(\boldsymbol{\theta}_t) \to 0$. A standard choice is
$$\kappa_t = \sqrt{2\log t}, \qquad t \geq 2.$$

Since $s_t(\boldsymbol{\theta}_t)$ decreases as $M_t$ grows, this ensures Equation 1 promotes exploration while maintaining vanishing variance. Empirically, $\kappa_t \equiv 1$ (EI) is sufficient, but the above choice yields fully rigorous convergence.

### B.4 THEOREM STATEMENT

Combining Equation 4 with $\lim_{t\to\infty} \rho_t = 0$, we obtain
$$\limsup_{t\to\infty} \mathbb{E}[f(\boldsymbol{\theta}_t)] \leq f^\star + \sigma,$$

and if $\sigma = 0$,
$$\lim_{t\to\infty} \mathbb{E}[f(\boldsymbol{\theta}_t)] = f^\star.$$

This proves Theorem 4.7.

## C EXPERIMENTAL AND ANALYSIS METHODOLOGY

### C.1 EXPERIMENTAL TESTBED SETUP

We run our synthesis experiments and classical processing tasks on our local computing cluster. The cluster consists of nodes with the AMD EPYC 7702P 64-core processor with x86_64 architecture and a 2.0 GHz clock. We spawn virtual machines (VMs) on these nodes consisting of 8 cores, 32 GB memory, and 32 GB storage for each of our experiments, providing more than sufficient resources for each experiment. The VMs are resource-bounded and not overprovisioned, ensuring that each experiment has exclusive access to the hardware resources assigned to it without any interference, which helps us provide accurate and consistent timing analysis.

We run all of our quantum experiments on the `ibm_nazca` quantum computer, a 127-qubit quantum computer with Eagle r3 architecture available via the IBM quantum cloud (Castelvecchi, 2017). The computer has a median one-qubit gate error of $3.341 \times 10^{-4}$, a median two-qubit gate error of $1.150 \times 10^{-2}$, and a median measurement operation error of $2.250 \times 10^{-2}$.

Table 1: List of software libraries used for the implementation and evaluation of CircuitTree.

| Software | Version | Software | Version |
|----------|---------|----------|---------|
| python | 3.12.3 | scikit-optimize | 0.10.2 |
| bqskit | 1.1.2 | qiskit-aer | 0.14.2 |
| qiskit | 1.1.0 | qiskit-ibm-runtime | 0.25.0 |
| SALib | 1.5.0 | scikit-learn | 1.5.0 |

## C.2 SOFTWARE FRAMEWORK IMPLEMENTATION

Table 1 provides a list of all the software used for the implementation and evaluation of CircuitTree. All libraries and packages are Python-based. We use `scikit-optimize` (Scikit-Optimize, 2024) to perform BO, with models from the `scikit-learn` library (Pedregosa et al., 2011) as surrogates. We use the `bqskit` library to run the state-of-the-art competitive synthesis framework (Group, 2021). We use the `qiskit` library (Aleksandrowicz et al., 2019) to create our circuit instruction sets, as it is developed by IBM to be compatible with the IBM quantum cloud and hardware. We use `qiskit_aer` to simulate the quantum circuits to get the circuit output during the optimization process. Our test circuits for evaluation metrics are taken directly or modified from `QASMBench` (Li et al., 2023), a benchmark suite of near-term circuits.

We use `qiskit_ibm_runtime` to interface with the IBM quantum cloud and run synthesized circuits on the `ibm_nazca` quantum computer. When transpiling the circuits to the `ibm_nazca` computer, we use the transpilation optimization level of 0 to eliminate the influence of confounding factors such as non-synthesis techniques for our analysis. We run all circuits with 10,000 shots by default unless specified otherwise. We run each circuit with each technique five times to account for statistical variabilities related to random seeds in the optimization models and show the mean and standard deviation for all the metrics.

We implement repeated layers of the ansatz shown in Fig. 1 for the implementation of CircuitTree. The ansatz consists of a collection of parameterized (optimizable) Ry and Rz gates, which can be used to implement a universal one-qubit quantum gate. This is followed by a collection of cascading two-qubit CX gates. These gates are organized to be compatible with a linear qubit-connection topology, which assumes that each qubit is at maximum connected to two other qubits and they are all connected in a sequence. This kind of sparse connectivity is common in superconducting quantum computing due to crosstalk and interference-related challenges faced by dense connectivity (Dumitrescu et al., 2020; Wright et al., 2019; Ravi et al., 2021).

Therefore, CircuitTree uses this sparse CX-gate format to avoid the insertion of additional SWAP gates (which have the noise footprint of three CX gates) to make non-connected qubits interact. These sequences of gates form one layer of the ansatz. Unless specified otherwise, we typically use 3 or 4 layers for our analysis, as that performs well empirically.

## C.3 RELEVANT ANALYSIS METRICS

**Total Variation Distance (TVD).** The TVD is a widely-used metric to measure the difference between two probability distributions (Oh et al., 2024; Clark & Thapliyal, 2024; Patel et al., 2022; Patel & Tiwari, 2021). For a quantum system of $n$ qubits with $2^n$ output states, the TVD between two probability distributions $P_1$ and $P_2$ over these $2^n$ states can be measured as $\text{TVD} = \frac{1}{2} \sum_{i=0}^{2^n-1} \left| p_1^i - p_2^i \right|$, where $p_1^i$ is the probability of observing state $i$ in distribution $P_1$ and $p_2^i$ is the probability of observing state $i$ in distribution $P_2$. This metric is used during our synthesis procedure and for technique evaluation to examine the quality of the results by comparing the output distribution of the synthesized circuit to the output distribution of the target circuit.

**Synthesis Time.** This is the overall time to run a given circuit synthesis method for any given quantum circuit. This metric is useful for comparing the optimization overhead (i.e., efficiency) of different circuit synthesis methods. We ensure that all synthesis methods are run on the same experimental testbed setup (described above) for a fair comparison.

**Circuit Depth.** This is the length of the critical path of a quantum circuit, i.e., the longest serial path traced from the first gate of the circuit to the last gate of the circuit. This metric is typically used as a proxy for circuit runtime. The lower the circuit depth, the better, as deeper circuits can lead to higher errors due to the decoherence of qubit states (Liu et al., 2020; Silver et al., 2023; Li et al., 2023; Wille et al., 2019).

**Number of Gates.** This refers to the circuit's total number of two-qubit CX gates. We only count the number of CX gates due to the fact that CX gates have an order of magnitude higher error rate than one-qubit gates and, thus, have a dominant impact on the overall output error (Tannu & Qureshi, 2019; Ravi et al., 2021; Patel et al., 2020b;a). As a result, the lower the total number of CX gates in the circuit, the lower the overall output error, making the number of gates an important metric.

### C.4 ALGORITHMS EVALUATED

We evaluate CircuitTree using algorithms with different characteristics, as described below.

**Randomly-Generated Quantum Circuits (RQC).** While designing CircuitTree, we used RQCs as synthesis targets to cover a variety of circuit behaviors. Randomly generated circuits play a crucial role in benchmarking and testing the capabilities of quantum processors. These circuits are used to assess the performance and reliability of design decisions by generating complex, unpredictable quantum states that stress the system's coherence and error rates (Boixo et al., 2018).

**Quantum State Preparation (QSP).** Amplitude embedding state preparation circuits are fundamental in quantum computing, enabling the encoding of classical data into quantum states by mapping data amplitudes to the amplitudes of quantum states (Schuld et al., 2019). Their significance lies in their ability to leverage quantum parallelism to represent large datasets and perform complex operations intractable for classical methods. However, implementing amplitude embedding circuits presents significant challenges, including the need to construct efficient quantum circuits that can precisely encode data while minimizing gate depth and errors – they are considerably deep circuits (Grover & Rudolph, 2002; Mitarai et al., 2018). We evaluate using the circuits for ten different randomly generated amplitude embedding states with real-valued coefficients, prepared using the `qiskit` state preparation algorithm (Javadi-Abhari et al., 2024; Shende et al., 2005). When applying CircuitTree to synthesize these circuits, our ansatz uses only Ry and CX gates to ensure that the coefficients of the state vector are real-valued.

**Variational Quantum Eigensolver (VQE).** VQE is a hybrid quantum-classical algorithm designed to find the ground state energy of a quantum system, making it particularly useful for quantum chemistry and materials science (Peruzzo et al., 2014). The significance of VQE lies in its ability to efficiently handle problems that are intractable for classical algorithms by exploiting quantum parallelism and entanglement. However, implementing VQE poses significant challenges due to its deep circuit, which includes mitigating noise and decoherence and efficiently optimizing synthesis parameters. We evaluate this circuit with a three-layer CircuitTree ansatz.

## D TRACE-BASED DISTANCE VERSUS TVD

Trace-based distances, such as Unitary Matrix Distance (UMD), can also be used as a cost function by setting $U_1$ to be the unitary operator representing the target circuit and $U_2$ to be the unitary operator representing the synthesized circuit. Prior synthesis works have typically used UMD-variant metrics as objective functions (Smith et al., 2021; Younis et al., 2021; Patel et al., 2022; Younis & Iancu, 2022; Weiden et al., 2022; Davis et al., 2020). This is due to their versatility in synthesizing for the inherent quantum computation, as opposed to just for the output distribution. However, we found that although surrogate modeling is capable of improving UMD for parameterized circuits, the improvement is slow, as demonstrated in the inset diagram in Fig. 7(a). The figure also illustrates how the TVD improves as the UMD is optimized, and the improvement in TVD is not as substantial as the TVD improvement observed by directly optimizing the TVD. Correspondingly, Fig. 7(b) shows that the output distribution fidelity of the circuit is substantially worse than what is obtained from using TVD. The output distribution using the TVD objective function closely resembles the target output distribution, while the output distribution using the UMD objective function is far off. Thus, the TVD objective function vastly outperforms the UMD objective function.

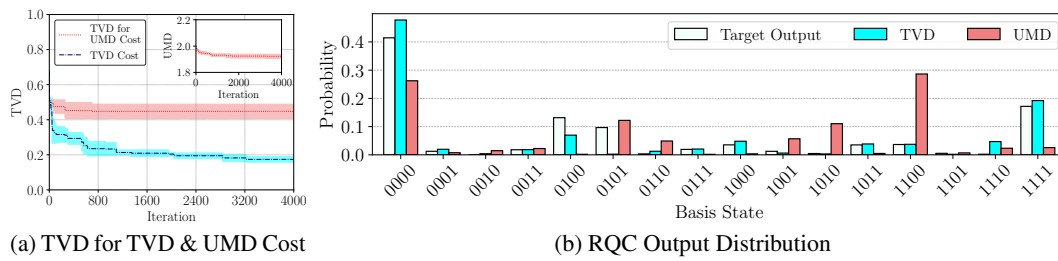

(a) TVD for TVD & UMD Cost        (b) RQC Output Distribution

Figure 7: (a) The TVD while optimizing a parameterized circuit to synthesize an RQC, using TVD or UMD as the cost (objective) function. The inset shows the convergence of UMD when used as a cost function. (b) The final distribution produced when using TVD cost is significantly closer to the target output distribution than that obtained using UMD cost for the same number of iterations. Both TVD and UMD runs were completed using Gradient Boosted Random Trees (GBRT) as the surrogate.

However, using the TVD for the objective function does not fully optimize the internal dynamics of the quantum circuit's computation. It instead focuses only on the classical-facing output distribution. This is an acceptable trade-off for CircuitTree because, similar to classical code, for all quantum circuits, the output is ultimately of consequence. It is not important to completely replicate the internal dynamics of a quantum program during the synthesis procedure as long as the generated output is correct. Using the UMD cost function attempts to replicate all the complex internal dynamics rather than the output distribution itself, and thus fails to achieve the high fidelity to the target output that using the TVD cost function achieves.

Therefore, CircuitTree uses TVD as the cost function, as TVD best corresponds to the goal of matching the target output distribution (see Fig. 7(b)). This also comes with some practical advantages, including relaxing the requirement to construct the unitary matrix representing the circuits, which can benefit our technique in terms of runtime and memory. One may hypothesize that using a multi-objective function that combines the TVD and UMD metrics in a weighted manner might provide the best results, as the TVD focuses on the output distribution and the UMD focuses on the internal quantum dynamics. However, due to the poor performance of the UMD metric, combining the two metrics only served to diminish the performance of the TVD metric, as it placed a strain on the optimizer. We, therefore, use only the TVD metric for our objective function.

# E    EXAMINING PARAMETRIC IMPORTANCE USING SOBOL ANALYSIS

We now analyze the impact of various gate parameters within each layer of the ansatz for the VQE circuit, ranked according to their Sobol indices in Fig. 8. Sobol indices are a measure of the sensitivity of the output of a model to its input parameters, indicating the relative importance of each parameter (Sobol, 1993). We compute these indices by sampling parameter values for each layer from a Sobol sequence. We then use CircuitTree-trained surrogate models to compute the objective function value at each of these points and estimate the Sobol indices using the SALib library (Herman & Usher, 2017). These indices are also accompanied by a 95% confidence interval (CI), which gives an uncertainty estimate for each index. Since each layer has a separate model that learns only the parameters within its own layer, this allows us to compare indices within that layer.

For our purposes, the Sobol indices are particularly useful in identifying which parameters most influence the output of a quantum circuit, potentially allowing for targeted optimizations and more efficient synthesis of the circuit. The analysis reveals that the Ry gates have a significantly higher impact on the circuit's performance compared to the Rz gates. Specifically, the Ry gate on qubit 0 in layer 0, the Ry gate on qubit 3 in layer 1, and the Ry gate on qubit 3 in layer 2 are identified as the most consequential gates. These gates are crucial in determining the overall fidelity of the synthesized VQE circuit. This highlights the importance of the Ry gates in creating and manipulating superposition, which is essential for the efficacy of the VQE algorithm, as opposed to the Rz gates, which introduce phase into the circuit.

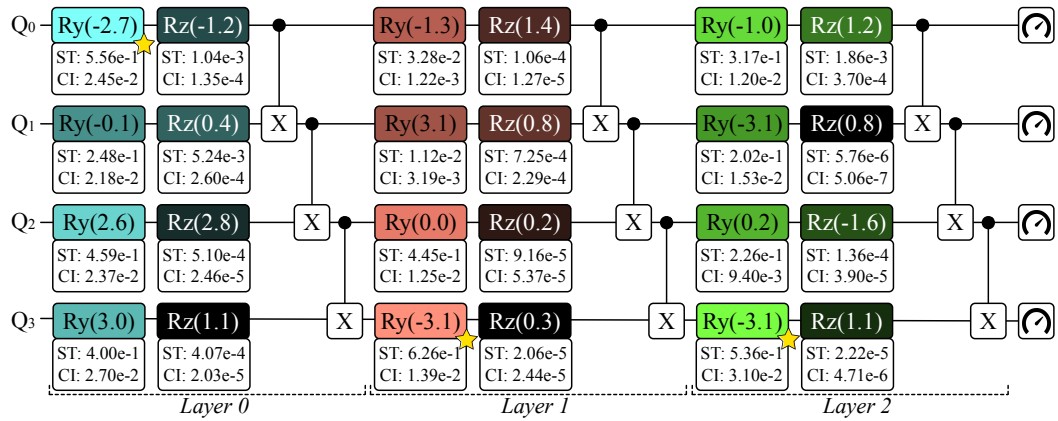

Figure 8: The impact of different parameters ranked according to Sobol indices for the VQE circuit. The numbers in the boxes show the final parameter values after CircuitTree's synthesis, while the boxes are colored according to their impact within the layer – the lighter the color, the higher the impact. The highest-impact gates are the Ry gate on qubit 0 in layer 0 and the Ry gate on qubit 3 in layers 1 and 2. Numerical values for each total-order Sobol index (ST) and 95% Confidence intervals (CI) are shown.

Interestingly, the impact of the gates does not correlate directly with their final parameter values. While the highest-ranked gates generally have larger absolute angle values, there are instances where less impactful gates have similar or even larger parameter values. For example, the Ry gate on qubit 2 in layer 1 has a final angle of ≈0.02, yet it holds a higher impact than the Ry gates on qubits 0 and 1 in layer 1, which have larger absolute angles. This suggests that the circuit's sensitivity to certain parameters is not trivially dependent on the magnitude of the gate angles, but also on their role and position within the circuit. This analysis highlights the potential to focus on specific gate parameters when optimizing quantum circuits in CircuitTree's layered approach. The differentiation in the impact of Ry and Rz gates also provides valuable insights into the optimization of quantum circuits, particularly in the context of variational algorithms like VQE.

## F    USE OF LARGE LANGUAGE MODELS (LLMs)

We acknowledge that LLMs such as ChatGPT and Grammarly AI were used in the writing of this work, specifically to polish it by correcting sentence mistakes and typographical errors.

## G    REPRODUCIBILITY STATEMENT

All code associated with this work has been fully open-sourced and is provided alongside the paper (attached in "Supplementary Materials"). The repository includes instructions, code files, scripts, and data to reproduce every experiment reported. By making our implementation publicly available, we aim to ensure transparency, facilitate independent verification of our results, and encourage community adoption and extension of our methods.

