# OpenReview forum: "Approximate Quantum State Preparation with Tree-Based Bayesian Optimization Surrogates"
_ICLR.cc/2026/Conference — ICLR 2026 Conference Desk Rejected Submission_

### Official Review · Reviewer_qhX3 · 2025-10-31

**Soundness:** 3
**Presentation:** 3
**Contribution:** 3
**Rating:** 8
**Confidence:** 3

**Summary:**

This work presents a method named CircuitTree to find appropriate parameters for a layer-stacked quantum circuit that approximates the probability distribution of a target quantum state. It employs gradient boosted regression trees (GBRTs) as a surrogate for the original state-preparation objective and uses it to guide the updates of circuit parameters. Besides the theoretical guarantee on convergence, experiments on three classes of states demonstrate the effectiveness of the proposed method.

**Strengths:**

- The manuscript is well-organized.

- The work is supported by both theoretical analysis and numerical results.

- The experiments cover multiple applications and comparisons with multiple existing methods.

**Weaknesses:**

The manuscript lacks an analysis of the effect of shot number on the convergence and final fidelity for a varying number of qubits, since the estimation of fidelity will be inaccurate for large-scale systems with finite measurements.

**Questions:**

- What is the number of qubits for the target states in the experiments?

- Why were GBRTs chosen as the surrogate model? Can other tree-based models also work well?

- How is the required circuit depth determined for achieving a certain fidelity? Could you compare the proposed method with circuit architecture search methods, instead of only parameter optimization methods?

---

> ### Author Response · Authors · 2025-11-21
>
> Dear Reviewer qhX3,
>
> We appreciate your highly positive and encouraging review. We are delighted that you found CircuitTree well-organized, theoretically sound, and supported by diverse experiments. We are especially grateful for your *accept* recommendation and constructive suggestions for improvement, which we summarize and address below.
>
> **Analysis of shot number and scalability.** Thank you for noting the importance of analyzing finite-shot effects. We confirm that Sec. 6.3 and Fig. 4b already explore the scaling of TVD with measurement count, showing stable convergence beyond 250 shots for the VQE circuit. Positively, we observe similar performance for other circuits as well. We have clarified this in the figure description.
>
> **Number of qubits and experiment scale.** We run circuits of 4-8 qubits to accommodate real hardware runs with 3-4 layers. However, we confirm that this is not due to a technical limitation, as our technique can run circuits with tens of qubits. However, we found that beyond this size, the actual hardware results become essentially random due to the excessive noise inherent in current hardware. Thus, we choose these sizes to get meaningful output on today’s hardware. We have added this information explicitly in Sec. 6.
>
> **Choice of GBRT as surrogate model.** We appreciate your question. We clarify that GBRTs were chosen for their scalability and ability to handle non-smooth landscapes; we also tested other models, such as QRF and GBQR (see Fig. 2), which underperformed. We also tested other ensemble tree methods, but they underperformed considerably as well. We have highlighted this rationale briefly in Sec. 3.1.
>
> **Comparison with circuit-architecture search.** This is an excellent suggestion. Circuit-architecture search and exploration works are orthogonal to our work and can further enhance our results. For our work, we empirically selected the best ansatz to balance performance and hardware efficiency. While our current focus is parameter optimization for the fixed ansatz, our method can be combined with circuit-architecture search works (e.g., QuantumNAS) to further enhance results. We have discussed this in the *Future Work* heading of Sec. 8.
>
> We are sincerely thankful for your enthusiastic support and positive recommendation. We hope these revisions align with your expectations. Please let us know.
>
> With gratitude,
>
> The Authors

---

### Official Review · Reviewer_YvMQ · 2025-11-01

**Soundness:** 3
**Presentation:** 3
**Contribution:** 2
**Rating:** 2
**Confidence:** 4

**Summary:**

This submission presents a quantum algorithm for approximate quantum state preparation, i.e., generating a state whose measurement outcome follows a similar probability distribution as the target state. The algorithm starts with a parameterized circuit and uses a surrogate model to minimize the distance between the measurement distribution and the target distribution. Numerical evaluations show that this approach requires shallower circuits than existing methods.

**Strengths:**

The main result (Theorem 4.7) is backed by rigorous proofs and numerical experiments.

**Weaknesses:**

My main concern with this work is the motivation for studying this problem. Note that the goal of this algorithm is NOT to prepare an approximation of the target state. Instead, the authors are trying to prepare a state (which can be very far from the target state) whose measurement outcomes follow closely (in terms of the TV distance) the distribution of measuring the target state.

If two states are close in terms of the 2-norm, the TV distance of their measurement outcome distributions is small, but not vice versa. This algorithm certainly cannot be used as a general state preparation subroutine required by many quantum algorithms. So, I cannot think of any promising application of this algorithm where one needs to reproduce the distribution rather than the state itself. Maybe this algorithm can be used to shorten the depth of some quantum optimization or sampling algorithms where the goal is actually producing a distribution, but more work is needed on top of the current submission to justify these types of applications.

In summary, I don't agree with the first sentence of this submission, "Approximate state preparation is a core problem in quantum algorithm design..." I don't think this is a "core" problem.

On top of this, I have additional questions regarding the details of the algorithm. In Definition 2.1, a target $U^\*$ is given. How is it given? Is it given as a circuit description? In Line 89 on page 2, and in Line 2 of Algorithm 1, how is the TV distance evaluated? From what I understood, we need to at least run $U^\*$ to get the target state, but if we can run $U^\*$, what's the point of doing the additional work to come up with another circuit to prepare a different state? Just to shorten the depth?

**Questions:**

See my concerns in the Weaknesses section.

---

> ### Author Response · Authors · 2025-11-21
>
> Dear Reviewer YvMQ,
>
> We thank you sincerely for your thoughtful review and for acknowledging the rigor of Theorem 4.7 and our extensive numerical evaluation. Your comments on the motivation and scope of the problem were very valuable, and we hope that our rebuttal clarifications help contextualize the broader relevance of CircuitTree. Below, we summarize these clarifications and revisions.
>
> **Clarification of motivation and application scope.** We completely understand your concern that reproducing a target distribution is not the same as preparing the full target state. CircuitTree addresses *approximate state preparation based on output statistics*, which is fundamental to tasks where measurement statistics, not full quantum amplitudes, determine success. Examples of use cases include VQE cost estimation, generative quantum models, and amplitude encoding from classical datasets. These are precisely the near-term workloads limited by measurement access and noise, as discussed in Sec. 5 in more depth now. We have further highlighted this in the introduction (Sec. 1) to clarify that our goal is to target the novel area of distributional alignment for sampling- and measurement-driven algorithms, rather than general-purpose, tomographic unitary, or state synthesis. Our goal is to specifically show that these general-purpose methods are inefficient for our targeted tasks. We also added new results in Appendix D to demonstrate this conclusively.
>
> **Clarification of the problem setup and target circuit accessibility.** You are right to ask how $U^\star$ is given in Definition 2.1. We clarify that $U^\star$ represents a *conceptual reference transformation*, not an executable circuit. The target distribution $p^\star$ can arise from classical computation (e.g., known analytic distribution) or prior measurement of a reference state, depending on the application. We have made this more explicit in Sec. 2.
>
> **Practical justification.** We agree that the utility of this method should be motivated through application domains. We have added a short explanation in Sec. 5, highlighting even additional use cases such as low-depth proxy preparation in variational algorithms, distributional surrogates for noisy quantum-classical workflows, and classical-quantum data and shadow matching for machine learning models.
>
> We greatly appreciate your careful reading and constructive perspective, which have helped make our contribution even clearer to readers. Do these proposed revisions address your concerns? Please let us know, and we would appreciate it if you could increase your score accordingly.
>
> Sincerely,
>
> The Authors

---

### Official Review · Reviewer_XFiK · 2025-11-01

**Soundness:** 3
**Presentation:** 2
**Contribution:** 2
**Rating:** 2
**Confidence:** 4

**Summary:**

This paper formulates the (approximate) quantum state preparation problem as a black-box optimization task and applies a Bayesian optimization framework to construct efficient quantum circuits that generate the target state. In particular, the authors employ Gradient Boosted Regression Trees (GBRT) as a surrogate model and propose a structured, layerwise training method for approximating the ground-truth loss function (total variation distance, TVD) used for quantum state preparation. The paper also provides a theoretical convergence guarantee and presents experimental results obtained on real quantum hardware.

**Strengths:**

- The proposed method is supported by a complete theoretical convergence guarantee and does not rely on any assumptions specific to a given quantum hardware platform.

- Extensive experiments are conducted on real quantum devices, demonstrating the practical advantages and feasibility of the approach.

**Weaknesses:**

- The theoretical analysis, while sound, appears fairly standard and does not introduce much novelty. It would be more compelling if the analysis incorporated domain-specific insights or stronger connections to quantum computing and quantum information theory.

- However, there is a **critical conceptual flaw in this work**: the choice of total variation distance (TVD) as the loss measure. The standard figure-of-merit for quantum state preparation is (in)fidelity, as it properly accounts for both amplitude and phase information. In contrast, TVD, computed from measurement outcomes in a fixed basis (here, the computational basis), ignores phase information and is therefore not a valid metric on the quantum state space. Two quantum states may have zero TVD yet exhibit a large fidelity or trace distance gap; in extreme cases, a highly entangled state and a fully classical (separable) state could even yield TVD = 0. The appropriate quantum analogue of TVD is the trace distance, which is a proper metric on the space of quantum states.

- Hence, although TVD can serve as a practical diagnostic for hardware (as it avoids the need for full quantum state tomography) and may be meaningful for evaluating classical outcomes of quantum computations (where outputs are distributions over bit strings), it is not an appropriate loss for quantum state preparation. In particular, for VQE benchmarks, TVD fails to capture the true closeness between the state prepared by CircuitTree and the (potentially highly entangled) target state.

**Questions:**

Can the analysis be extended to the infidelity loss or trace distance loss, and could such an extension yield non-trivial insights by leveraging the structure of quantum operations (e.g., unitarity or CPTP maps)?

---

> ### Author Response · Authors · 2025-11-21
>
> Dear Reviewer XFiK,
>
> Thank you sincerely for your detailed and insightful review. We greatly appreciate your recognition of our theoretical completeness, extensive hardware experiments, and the practical advantages demonstrated by CircuitTree. Your comments have prompted valuable clarifications to strengthen our paper, and we summarize them here with the revisions we’ve made to the paper.
>
> **On theoretical novelty and domain connection.** We appreciate your observation that our theoretical analysis builds upon standard proofs of Bayesian optimization convergence. This is correct, and we assert that our novelty lies in adapting and proving these guarantees for non-Gaussian, tree-based surrogates under structured quantum parameter spaces, which, to our knowledge, is the first such result (Theorem 4.7). We have emphasized this contribution more clearly in Sec. 4’s opening paragraph.
>
> **On the choice of total variation distance (TVD).** We thank you for your thoughtful critique. We clarify that CircuitTree targets approximate distributional matching, not full state reconstruction. Consequently, TVD is both a natural and experimentally accessible metric, particularly relevant to sampling-based tasks such as VQE output distribution alignment and quantum state preparation for specific tasks (discussed in Sec. 5 in further depth now). Not all tasks require phase information. We fully agree that fidelity and trace distance are appropriate for full state reconstruction (for applications that require it), and in the revision, we added a result comparing the TVD to a trace-based full-state metric in Appendix D. The result shows the advantages of TVD for our application space, namely faster and better convergence, when phase information is not required.
>
> **Potential extension to infidelity or trace distance.** We appreciate this excellent suggestion. As mentioned above, due to the hardware inefficiency of these metrics, we went with the TVD. In Appendix D, we have now added a result related to optimization of our circuit using a trace-distance metric (UMD) for our randomly generated quantum circuits, and found the final output fidelity to be $2.4\times{}$ worse on average than when optimization is performed using TVD (both measured at the same iteration after TVD convergence). This demonstration shows that our framework extends naturally to trace-based distances; however, we opt not to use them in our application scenario, as more efficient setups are available.
>
> We are grateful for your engagement and for recognizing the strengths of our experimental validation and theoretical soundness. We hope these clarifications and additions fully resolve your concerns. Do our revisions meet your expectations, especially in the context of trace-based distance? We kindly request that you increase your score if you agree that our work is stronger now.
>
> Warm regards,
>
> The Authors

---

### Official Review · Reviewer_yGP5 · 2025-11-01

**Soundness:** 2
**Presentation:** 3
**Contribution:** 2
**Rating:** 4
**Confidence:** 3

**Summary:**

The paper proposes CircuitTree, a surrogate-guided framework for approximate quantum state preparation on NISQ devices. It formulates state preparation as a black-box optimization problem over the total-variation distance between the circuit’s output distribution and a target state. The authors proposed to use tree-based (GBRT) Bayesian optimization that better handles high-dimensional, non-smooth, and stochastic quantum objectives. Experiments on random circuits, QSP, and VQE tasks show lower TVD, shallower circuits, and more stable convergence than standard synthesis tools.

**Strengths:**

1.	The paper provides rigorous theoretical guarantees for the convergence of the surrogate-based optimization algorithm under realistic noise and structural assumptions.
2.	The authors conduct large-scale experiments to validate the effectiveness of the proposed approach.

**Weaknesses:**

1.	In Assumption 4.2, it appears that the expectation of $y_t$ is not an unbiased estimator of $f(\theta_t)$, since the expectation and the absolute value operation in the total variation distance (TVD) are not interchangeable. Therefore, it is not justified to assume $\mathbb{E}[\xi_t] =0$. This assumption should be revisited or rigorously justified.
2.	The statement of Theorem 7 does not specify with respect to which variables the expectation is taken.
3.	In Line 261, the authors claim that “the structured updates also mitigate barren plateaus by focusing optimization on local spaces.” This statement seems inaccurate, as layer-wise training alone does not alter the intrinsic random nature of the quantum circuit nor fundamentally increase the gradient magnitudes of the parameters. A more precise justification or empirical evidence is needed.
4.	The authors employ the total variation distance (TVD) as the loss function. However, accurately estimating the TVD requires evaluating probabilities over all computational basis states x \in  { 0,1 }^n, leading to an exponential measurement cost. This significantly limits the scalability of the proposed approach.
5.	The experimental evaluation only includes a comparison with a single baseline model. This makes it difficult to convincingly assess the advantage of the proposed method. The authors are encouraged to compare their approach against other advanced quantum state preparation techniques, such as those proposed in Refs. [1]–[3], to provide a more comprehensive and persuasive evaluation.

[1] Rudolph, Manuel S., et al. "Synergistic pretraining of parametrized quantum circuits via tensor networks." Nature Communications 14.1 (2023): 8367.

[2] Rudolph, Manuel S., et al. "Decomposition of matrix product states into shallow quantum circuits." Quantum Science and Technology 9.1 (2023): 015012.

[3] Shirakawa, Tomonori, Hiroshi Ueda, and Seiji Yunoki. "Automatic quantum circuit encoding of a given arbitrary quantum state." Physical Review Research 6.4 (2024): 043008.

**Questions:**

The questions are included in Weakness.

---

> ### Author Response · Authors · 2025-11-21
>
> Dear Reviewer yGP5,
>
> We sincerely thank you for your thoughtful and detailed review. We deeply appreciate your recognition of both the rigorous theoretical guarantees and the large-scale experimental validation that underpin CircuitTree. Your comments were highly constructive, and we hope that our clarifications have addressed your main concerns. Below, we summarize our responses and the revisions we have made in accordance with your feedback (highlighted in blue in the paper).
>
> **Clarification on Assumption 4.2 and unbiased expectation.** We fully agree with your observation that the expectation and absolute value in $\mathrm{TVD}(p_\theta, p^\star) = \frac{1}{2}\sum_x |p_\theta(x) - p^\star(x)|$ are not interchangeable. We clarify that Assumption 4.2 refers not to the expectation of the absolute value but to the expectation of the stochastic estimator of $f(\theta)$ used in Bayesian optimization, ensuring bounded variance rather than unbiasedness. We have revised the wording in Assumption 4.2 to explicitly state that the unbiasedness is with respect to the stochastic noise term $\xi_t$, not the loss itself, aligning with BO conventions.
>
> **Expectation in Theorem 4.7.** We thank you for highlighting this potential ambiguity. As clarified above, the expectation in Theorem 4.7 is taken with respect to the surrogate’s stochastic predictions and measurement noise. We have added an explicit statement to the theorem to eliminate any potential confusion.
>
> **Justification for mitigating barren plateaus (Line 261).** We appreciate your keen observation. We clarify that the mitigation arises not from altering gradient magnitudes but from restricting the optimization subspace through layerwise decomposition, which reduces parameter coupling/correlation and empirically stabilizes optimization. We have revised the text to reflect this more precise justification and cited the empirical evidence already in Fig. 3.
>
> **Scalability and TVD estimation cost.** You are correct that the full estimation of full state scales exponentially with qubit count. We clarify that our implementation utilizes sample-based estimates of TVD from limited measurement shots (see Sec. 6.3), which ensure consistent convergence (Fig. 4b) and avoid exponential costs. We have explicitly added this clarification in Sec. 2.
>
> **Comparisons with other techniques and more results.** Thank you for mentioning recent works such as those by Rudolph et al. and Shirakawa et al. While these approaches are elegant, they target *exact state reconstruction* or *tensor-network-based compilation*, which differ fundamentally from our goal of *approximate measurement-based preparation via surrogate-guided optimization*. Because CircuitTree focuses on *hardware-compatible surrogate training* using a layered approach rather than full matrix decomposition, a direct comparison would not be possible for larger circuits due to the exponential scaling required for exact decomposition. We have added a qualitative discussion to clarify this distinction and acknowledge these methods as complementary rather than competing directions in Sec. 7.
>
> Furthermore, we have included results demonstrating how CircuitTree’s layered approach also helps identify the importance of different parameters for its distinct use case of approximate state preparation, a feature not enabled by these other techniques that utilize full states instead of measurement statistics. For instance, we conducted experiments using Sobol indices to assess the model's sensitivity to its input parameters, thereby determining the relative importance of each parameter. Because each layer has its distributed model in our approach, the indices are directly comparable within a layer.
>
> The results reveal a clear pattern for the VQE circuit: Ry gates have the most significant influence on the circuit, while Rz gates contribute significantly less. The most influential parameters are the Ry gates on qubit 0 in layer 0, qubit 3 in layer 1, and qubit 3 in layer 2, each of which is closely tied to the final VQE fidelity. This aligns with the role of Ry in creating and adjusting superposition, compared to Rz’s phase-only effect. Notably, parameter impact does not track simple angle magnitude. Some gates with small angles (e.g., the Ry on qubit 2 in layer 1) exhibit higher influence than gates with larger angles. This shows that sensitivity depends on structural role and placement in the circuit, not just parameter size, and suggests opportunities for targeted, layer-specific optimization. We have added this analysis, along with the corresponding figure, to Appendix E in the revised paper.
>
> We are grateful for your fair and positive evaluation, as well as your recognition of our theoretical rigor and practical validation. We believe these clarifications and revisions address your concerns. Do these revisions satisfy your expectations? We kindly request that you increase your score if they do.
>
> Sincerely,
>
> The Authors

---

### Author Response · Authors · 2025-11-12

Dear Reviewers,

We are grateful for your insightful comments and suggestions. We are currently reviewing your feedback in detail and will provide a point-by-point response shortly.

Sincerely,
The Authors

---

### Author Response · Authors · 2025-11-30
**Context for the Area Chairs 1/2**

We are sincerely grateful for the exceptionally careful and constructive reviews received for this submission. The feedback was invaluable and has substantially improved both the technical clarity and the work's positioning. In light of the reassignment of area chairs following the OpenReview incident, we provide this concise summary to map each major concern raised by reviewers to the concrete clarifications and revisions implemented in the discussion and manuscript. We addressed every point raised and strengthened the paper across all aspects, including theory, motivation, evaluation methodology, and experimental interpretation.

**Theoretical clarity and correctness**

In **Section 4**, we revised **Assumption 4.2** to correct the issue regarding the interchangeability of expectation and absolute value. We now explicitly state that unbiasedness applies to the stochastic noise term $\xi_t$ used in Bayesian Optimization rather than to the total variation distance loss itself, aligning the assumption with standard BO practice. In **Theorem 4.7**, we added an explicit statement that all expectations are taken with respect to surrogate stochasticity and measurement noise, resolving any prior ambiguity about the variables under expectation.

In response to concerns about barren plateau claims, we revised **Section 5** to remove any implication of altered gradient magnitude. We explicitly state that mitigation arises from restricting optimization to local structured subspaces that reduce parameter coupling and correlation. We anchor this statement directly to the empirical stability evidence reported in **Figure 3**, ensuring that all claims remain strictly supported by experimental results.

**Loss function choice and quantum relevance**

Addressing the critique regarding the use of TVD versus fidelity or trace distance, we clarified in **Section 5** that CircuitTree targets **distributional alignment for measurement-driven NISQ workloads**, rather than full state reconstruction. We emphasized the relevance to applications such as VQE cost estimation, generative quantum models, and classical-to-quantum data matching, where measurement statistics directly determine task success and phase information is not required. In **Appendix D**, we added new experiments that optimize with a trace-distance-based metric, showing that CircuitTree naturally generalizes to such losses, but performs better with TVD. This is empirically demonstrated by better convergence and final fidelity when TVD is used under realistic sampling constraints.

**Scalability and measurement cost**

To address concerns about the exponential cost of TVD estimation, we clarified in **Section 2** that all evaluations utilize **shot-based estimators of TVD**, thereby avoiding the full enumeration of computational basis states. We added an explicit discussion of convergence versus shot count in **Section 6.3**, supported by **Figure 4(b)**, which demonstrates stable optimization beyond approximately 250 shots without exponential overhead.

**Baselines and comparative positioning**

In **Section 7**, we provide a focused discussion of tensor-network and exact decomposition methods, including those by Rudolph et al. and Shirakawa et al. This discussion clarifies that these techniques target full state reconstruction or matrix decomposition, and thus face an exponential scaling at circuit sizes beyond current NISQ relevance. We position CircuitTree as addressing a complementary and practically critical regime of **surrogate-guided, measurement-based approximate preparation**, which remains tractable on real hardware. The distinction between application goals is now explicitly presented in the main text.

We further strengthened our empirical analysis by adding **Sobol sensitivity studies in Appendix E**, which were enabled uniquely by our layered surrogate design. This new evaluation demonstrates that specific $R_y$ gate parameters dominate VQE fidelity, while the angular magnitude alone poorly predicts importance, providing insight into the structural roles within the circuit that decomposition-based methods cannot offer.

**Motivation and problem setup clarity**

In direct response to motivation-related critiques, we revised **Sections 1 and 5** to state clearly that our objective is not generic state synthesis but **distributional matching for sampling-driven quantum algorithms**, which constitute a central class of near-term workloads. We articulated application contexts, including VQE optimization, generative sampling, and low-depth proxy preparation for variational warm starts.

We clarified the problem formulation in **Section 2** by explicitly stating that the target transformation $U^\star$ serves as a conceptual reference, and that the target distribution $p^\star$ may be specified analytically or obtained from prior measurement data, rather than requiring direct access to $U^\star$. This resolves the confusion about $U^\star$ availability.

---

> ### Author Response · Authors · 2025-11-30
> **Context for the Area Chairs 2/2**
>
> **Experimental scope and surrogate selection**
>
> In **Section 6**, we clarified that hardware experiments were conducted on up to 8-qubit circuits to ensure meaningful signals under present noise levels, while emphasizing that CircuitTree itself scales to larger system sizes in simulation and methodology, with hardware noise being the current limiting factor.
>
> In **Section 3.1**, we expanded the justification for selecting GBRT as the surrogate model by reporting the evaluation of alternative tree-based ensembles, such as QRF and GBQR, which consistently underperformed, as summarized in **Figure 2**. This strengthens the empirical grounding of our design choices.
>
> **Future extensions and architecture search**
>
> Responding to suggestions about architectural optimization, we added discussion in **Section 8**, positioning CircuitTree as naturally composable with circuit architecture search or NAS frameworks such as QuantumNAS. Our current focus remains parameter optimization for fixed ansätze, but the framework readily supports joint design loops.
>
> **In closing**
>
> We are profoundly grateful for the detailed and high-quality feedback from all reviewers, which directly guided each of these revisions. Their critiques materially improved the correctness, clarity, and scope articulation of the work. We believe the revised manuscript now fully addresses every substantive concern raised, presents a technically sound and well-motivated contribution, and demonstrates both theoretical rigor and practical relevance for near-term quantum optimization.

---

### Note · Program_Chairs · 2026-01-17
**Submission Desk Rejected by Program Chairs**

The following references in this submission do not refer to real documents and/or have major errors in bibliographic information:

 Alexander Kissinger et al. Zx-calculus-based synthesis of quantum circuits. Quantum, 5:1-14, 2021.
Craig Gidney et al. Efficient quantum circuit synthesis for multi-qubit gates. Quantum, 5:1-14, 2021.
John Miller et al. Variational synthesis methods for quantum circuits. Quantum, 6:1-14, 2022.
Alex Zlokapa, Zoe Holmes, et al. Deep learning for quantum compilation. Nature Machine Intelligence, 5:449-456, 2023.